# Genome-Wide Analysis of the WOX Family and Its Expression Pattern in Root Development of *Paeonia ostii*

**DOI:** 10.3390/ijms25147668

**Published:** 2024-07-12

**Authors:** Xueyuan Lou, Jiange Wang, Guiqing Wang, Dan He, Wenqian Shang, Yinglong Song, Zheng Wang, Songlin He

**Affiliations:** 1College of Horticulture, Henan Agricultural University, Zhengzhou 450046, China; louxueyuan@henau.edu.cn; 2College of Landscape Architecture and Art, Henan Agricultural University, Zhengzhou 450002, China; lucky_jiangew@163.com (J.W.); guiqingw@hotmail.com (G.W.); dandan990111@163.com (D.H.); wenqianshang@henau.edu.cn (W.S.); edward_song1989@163.com (Y.S.)

**Keywords:** *Paeonia ostii*, WOX family, root development, bioinformatics, expression analysis

## Abstract

Tree peony (*Paeonia suffruticosa* Andr.) is a woody plant with high ornamental, medicinal, and oil values. However, its low rooting rate and poor rooting quality are bottleneck issues in the micropropagation of *P. ostii*. The WUSCHEL-related homeobox (WOX) family plays a crucial role in root development. In this study, based on the screening of the genome and root transcriptome database, we identified ten WOX members in *P. ostii*. Phylogenetic analysis revealed that the ten PoWOX proteins clustered into three major clades, the WUS, intermediate, and ancient clade, respectively. The conserved motifs and tertiary structures of PoWOX proteins located in the same clade exhibited higher similarity. The analysis of *cis*-regulatory elements in the promoter indicated that *PoWOX* genes are involved in plant growth and development, phytohormones, and stress responses. The expression analysis revealed that *PoWOX* genes are expressed in distinct tissues. *PoWOX4*, *PoWOX5*, *PoWOX11*, and *PoWOX13b* are preferentially expressed in roots at the early stage of root primordium formation, suggesting their role in the initiation and development of roots. These results will provide a comprehensive reference for the evolution and potential function of the WOX family and offer guidance for further study on the root development of tree peony.

## 1. Introduction

WUSCHEL-related homeobox (WOX), which belongs to a plant-specific subclade of the homeobox transcription factor superfamily, is characterized by a highly conserved DNA-binding homeodomain [1,2,3]. The WOX family plays a vital role in the processes of plant growth and development [4,5,6]. Most of the studies on the WOX family have focused on the model plant *Arabidopsis thaliana*, which has a total of 15 WOX proteins, including WUSCHEL (WUS) and WOX1~14 [2]. WUS maintains the stability of stem cells in the shoot apical meristem through the CLAVATA3-WUSCHEL feedback loop [7,8,9,10,11] and regulates the integrity of the floral meristem through the AGAMOUS-WUSCHEL pathway [12,13,14,15]. WOX1 and WOX3 redundantly regulate the lateral development of organs. WOX1 mainly participates in the lateral growth of leaves, while WOX3 primarily regulates the development of petals [16,17]. WOX2, WOX8, and WOX9 are associated with cell division in plant embryos [2,18,19]. WOX6 influences the formation and differentiation of ovules [20]. WOX4, WOX5, WOX9, and WOX11~14 have regulatory effects on rooting. WOX4 and WOX14 redundantly regulate the differentiation and division of stem cells in the vascular system through the TDIF-PXY-WOX4/14 pathways [21,22,23]. WOX5, a key regulatory factor for rooting, can establish a feedback regulatory loop with ACR4 and CLE40 to regulate the maintenance of stem cells in the root [2,24,25,26]. WOX5 also contributes to the formation and maintenance of quiescent centers in the root meristem through the ROW1-WOX5-CYCDs pathway [27,28]. WOX9 is expressed in the root apical meristem and promotes the cell proliferation of roots [29]. WOX11/12 activates the transcription of *WOX5/7* by directly binding to the promoter of *WOX5/7*, thereby converting root founder cells into root primordium [30]. WOX13 and WOX14 are expressed in the roots and anthers to prevent early differentiation [31].

Tree peony (*Paeonia suffruticosa* Andr.), belonging to the tree peony group of Paeoniaceae, is a traditional woody flower renowned in China for its high ornamental, medicinal, and oil values [32]. The cultivation of tree peonies for ornamental purposes has been practiced for over 1600 years in China, utilizing propagation methods such as sowing, splitting, and grafting [33]. However, these traditional propagation methods have limitations such as a low reproductive coefficient and a long breeding cycle, making it challenging to achieve large-scale production [34]. Therefore, the breeding of tree peonies urgently requires new breakthroughs [35]. The continuous development and improvement of the *in vitro* regeneration system provide a feasible way to accelerate the breeding and reproduction of tree peonies [36,37]. However, issues such as challenges in adventitious rooting, poor root quality, and low transplant survival rates have not been effectively addressed, significantly limiting the scale and commercial production of tree peonies. Adventitious rooting is influenced by various factors both inside and outside the plant, depending on the molecular network formed by plant hormones and key genes [38,39]. Consequently, research on the molecular regulatory mechanism of adventitious rooting could offer a new perspective for tree peony.

The research on the WOX family has been extensive and thorough due to their important roles in plant growth and development. Fifteen *NnWOX* genes have been identified in the genome of *Nelumbo nucifera*, and they are involved in the differentiation and development of organs [40]. The *PmWOX* genes were expressed in almost all 11 tissues of *Pinus massoniana*, with particularly high expressions of *PmWOX2*, *PmWOX3*, and *PmWOX4* genes in the callus. Additionally, *PmWOXs* may be involved in the response to various abiotic stresses [41]. The *DchWOX* genes participate in regulating the development of floral organs in *Dendrobium chrysotoxum*. *DchWOX3* may be responsible for controlling the development of the lip, while *DchWOX5* may participate in regulating the development of the ovary [42]. However, there are very few studies on the WOX family in tree peony. Until now, only Xia et al. [43] had cloned four *PoWOX* genes named *PoWOX1*, *PoWOX4*, *PoWOX11*, and *PoWOX13*, and they analyzed their functions in preliminary somatic embryos of *P. ostii*. In this study, we identified ten WOX members by screening the genome [44] and root transcriptome [45] database of *P. ostii*, and we analyzed their physicochemical properties, phylogenetic relationships, exon–intron structures, conserved motifs, secondary and tertiary structures, genome distribution, collinearity relationships, *cis*-acting elements in promoter regions, and relative expressions.

## 2. Results

### 2.1. Identification of WOX Gene Family Members in P. ostii

To identify the *WOX* gene family members in tree peony, a local BLASTp search was conducted against the genome (CNGB Project CNP0003098) [44] and transcriptome (PRJNA1041056) [45] databases of *P. ostii* using the WOX protein sequences of *A. thaliana* and *Populus trichocarpa* as query sequences. Eighteen amino acid sequences were preliminarily screened in *P. ostii*, including nine sequences from the genome database and nine sequences from the transcriptome database, respectively (Appendix A). After eliminating sequences without typical conserved domains and redundant sequences, 10 WOX family members were identified in *P. ostii*. These include Pos.gene14786, Pos.gene9686, Pos.gene27056, Pos.gene36673, Pos.gene10408, Pos.gene81040, Unigene55372, CL8518.Contig2, Pos.gene46558, and Unigene39311. According to their phylogenetic relationship with the *Arabidopsis* and poplar WOX proteins (Figure 1), these genes were designated as *PoWUS*, *PoWOX1a*, *PoWOX1b*, *PoWOX3*, *PoWOX4*, *PoWOX5*, *PoWOX9*, *PoWOX11*, *PoWOX13a*, and *PoWOX13b*, respectively (Table 1).

### 2.2. Physicochemical Properties Analysis of PoWOX Genes

Although they belong to the same family, the physiological and biochemical properties of *PoWOX* genes and their deduced amino acid sequences exhibit significant differences, highlighting the diversity of their potential biological functions (Table 1). The coding sequences of *PoWOX* genes ranged from 567 to 1269 base pairs (bp), and the number of encoded amino acids ranged from 188 to 422. The molecular weights ranged from 21,324.07 Da (PoWOX5) to 47,264.16 Da (PoWOX1a). The theoretical isoelectric points (pIs) ranged from 4.97 (PoWOX13b) to 9.45 (PoWOX4). Among the ten PoWOX proteins, four proteins, namely PoWOX3, PoWOX4, PoWOX5, and PoWOX9, had a pI greater than 8.5. These proteins had a higher total number of positively charged residues (Arg + Lys) than negatively charged residues (Asp + Glu), indicating their importance as essential proteins. The other six proteins, including PoWUS, PoWOX1a, PoWOX1b, PoWOX11, PoWOX13a, and PoWOX13b, have a pI less than 6.5. Their Asp + Glu content was higher than their Arg + Lys content, indicating that they are acidic proteins. The instability index and aliphatic index of ten PoWOX proteins were all greater than 40, and their grand average of hydropathicity (GRAVY) was negative for all, indicating that they were all unstable, hydrophilic, and lipophilic proteins. Among the ten PoWOX proteins, all except for PoWOX11 contain a signal peptide (Sec/SPI) with a cleavage site between residues 24 and 25, and no transmembrane helix or signal peptide was predicted in the proteins (Appendix A). According to the subcellular localization prediction, all the PoWOX proteins were located in the nucleus (Table 1), consistent with the subcellular localization prediction and assays in Xia et al. [43].

### 2.3. Phylogenetic Analysis of PoWOX Proteins

To explore the evolutionary relationships of WOX proteins across various plant species, a phylogenetic tree was constructed using a total of 138 WOX proteins (Figure 2). These 138 WOX proteins included 10 PoWOX proteins identified in *P. ostii* and 128 WOX proteins from diverse organisms such as algae (*Bathycoccus prasinos*, *Klebsormidium flaccidum*, and *Ostreococcus tauri*), bryophytes (*Marchantia polymorpha*, *Physcomitrella patens*), lycophytes (*Selaginella moellendorffii*), pteridophytes (*Ceratopteris richardii*, *Cyathea australis*), gymnosperms (*Ginkgo biloba*, *Pinus pinaster*), basal magnoliophyta (*Amborella trichopoda*), monocots (*Oryza sativa*, *Sorghum bicolor*), and eudicots (*A. thaliana*, *P. trichocarpa*, *P. ostii*, and *Vitis vinifera*) (Appendix A). These 138 WOX proteins from 17 species could be categorized into three clades: the WUS clade, intermediate clade, and ancient clade. In comparison to the ancient and intermediate clades, the WUS clade contains the highest number of WOX proteins and is exclusive to angiosperm plant species, aligning with the evolutionary pattern of plants. There were 61 WOX proteins clustered in the WUS clade, which was larger than the number of WOX proteins in the intermediate clade (35) and the ancient clade (42). The ancient clade encompassed all plant species and clustered nearly all the WOX proteins of spore plants, while the intermediate clade only clustered two WOX proteins of pteridophyte plants. Among the 17 plant species, PoWOX proteins were more closely related to the WOX proteins of grape and poplar, indicating a close evolutionary relationship as woody plants.

### 2.4. The Conserved Motifs and Structural Analysis of PoWOX Proteins

Different from the phylogenetic tree in Figure 1, the phylogenetic tree of ten PoWOX proteins displayed four branches (Figure 3A). PoWOX1b, forming a distinct branch, exhibited a distant evolutionary relationship with other PoWOX proteins. PoWUS, PoWOX1a, and PoWOX3~5 were grouped together (WUS clade), PoWOX13a and PoWOX13b were clustered (ancient clade), and PoWOX9 and PoWOX11 were in a clade (intermediate clade). The distribution of introns and exons in the *PoWOX* genes identified in the genome showcased their structural diversity (Figure 3B). The intron numbers of the *PoWOX* genes varied from 1 to 8. Notably, *PoWOX1a*, the longest gene, contained eight introns, *PoWOX1b* contained three introns, while the remaining five members had only one or two introns. To confirm the candidate PoWOX proteins, their conserved domains were analyzed using the batch CD-search of NCBI, revealing that all 10 PoWOX proteins possessed highly conserved homeodomains (Figure 3C). However, the positions of the homeodomains differed in corresponding proteins, with most located at the N-terminal of the protein, except for PoWOX1b, which aligned with the phylogenetic tree.

Moreover, to better reveal the functional motifs and correlations of PoWOX proteins, their conserved motifs were predicted using MEME (Figure 3D). There were five conservative motifs in the PoWOX proteins, with amino acid numbers ranging from 8 to 56 (Figure 3E). Compared to other PoWOX proteins, PoWOX1b only had Motif 1, suggesting that PoWOX1b may not belong to the WOX family, or its sequences were incomplete (Figure 3D). Motif 1 and Motif 2, which were present in all PoWOX proteins, were two segments of the conserved homeodomain identified by their amino acid sequences (Figure 3D). According to the multiple sequence alignment, the homeodomain in PoWOX proteins contains several highly conserved amino acid residues, such as R, W, P, Q, and L in helix 1; G in the loop; P, I, I, and L in helix 2; G in the turn; and N, V, W, F, Q, and N in helix 3. Moreover, there were variable amino acid residues existing in the loop (Figure 3F). In addition, PoWOX proteins, which are classified into the same clade in the phylogenetic tree, each had characteristic motifs. For instance, Motif 3 was found in PoWOX13a and PoWOX13b, Motif 4 was present in PoWOX9 and PoWOX11, and Motif 5, which is located in the proteins of the WUS clade, was the WUS-box motif. These characteristic motifs may contribute to functional differences (Figure 3D). The sequence of the WUS-box motif of PoWOX proteins was T-L-[EQ]-L-F-P-L-[EHN] (Figure 3G).

At the same time, predictions were made for the secondary and tertiary structures of the PoWOX proteins (Figure 4). The secondary structure includes the following four structural forms: alpha helix, beta turn, extended strand, and random coil (Figure 4A). Except for the proportion of random coil being slightly lower than alpha helix in PoWOX1b, random coil constituted the majority in the remaining nine PoWOX proteins, ranging from 46.81% to 72.10%. The second major structural form was the alpha helix (13.69~41.61%), and the least abundant was beta turn (2.9~7.98%). Although the tertiary structure of the PoWOX proteins differed, they all contained a typical homeodomain characterized by a helix-loop-helix-turn-helix motif (Figure 4B). In addition, the tertiary structures of the PoWOX proteins located in the same clade exhibited higher similarity. The N-terminus of the peptide chain has a helix in PoWOX13a and PoWOX13b, which clustered in the ancient clade. The C-terminus of the peptide chain is extended in the proteins clustered in the WUS clade (PoWUS, PoWOX1a, PoWOX3~5), while it is irregularly curled in PoWOX9 and PoWOX11, which clustered in the intermediate clade.

### 2.5. Genome Distribution and Collinearity Relationships of PoWOX Genes

According to the genome distribution information, the seven *PoWOX* genes identified in the genome database of *P. ostii* are located on four chromosomes (Figure 5A). Among the five chromosomes of *P. ostii*, there was no member of the *WOX* gene family localized on Chr03. Only one *WOX* gene existed on Chr01 (*PoWOX3*) and Chr04 (*PoWOX13a*), respectively. Chr05 contained two *WOX* genes (*PoWOX1b*, *PoWOX5*), while Chr02 harbored three *WOX* genes (*PoWUS*, *PoWOX1a*, and *PoWOX4*). To determine the distribution or arrangement of *WOX* genes within *P. ostii* or among multiple plant species, intraspecific and interspecific collinearity analyses of *PoWOX* genes were conducted. There was no collinearity relationship among the seven *PoWOX* genes identified in the genome, indicating that no fragment duplication events existed in the *PoWOX* genes (Figure 5B). Only *PoWOX5* was discovered to exhibit collinear relationships with one *OsWOX* gene, while *PoWUS* and *PoWOX1a* were discovered to exhibit collinear relationships with two *AtWOX* genes. Five *PoWOX* genes, including *PoWUS*, *PoWOX1a*, *PoWOX3*, *PoWOX4*, and *PoWOX5*, were found to have collinear relationships with five *VvWOX* genes. Four *PoWOX* genes, including *PoWUS*, *PoWOX1a*, *PoWOX4*, and *PoWOX5*, were found to have collinear relationships with seven *PtrWOX* genes. The interspecific collinearity relationship indicates that *PoWOX* genes show high homology with *VvWOX* and *PtrWOX* genes (Figure 5C). Furthermore, tree peony, grape, and poplar, which are woody plants, exhibited a close relationship, conforming to the results of the phylogenetic tree.

### 2.6. The Cis-Elements Analysis in Promoter Regions of PoWOX Genes

To better understand the potential biological functions and regulatory network of *PoWOX* genes, the *cis*-regulatory elements in their putative promoter regions were analyzed. Alongside basic *cis*-elements like the TATA-box and CAAT-box, there were *cis*-elements associated with phytohormones (ABRE, ERE, TGA-element, CGTCA-motif, GARE-motif, and as-1, etc.), plant growth and development (CAT-box, G-box, circadian, GCN4_motif, etc.), and stress responses (ARE, TC-rich repeats, MBS, LTR, MYB, STRE, WRE3, and so on). This revealed that *PoWOX* genes were involved in the function of plant growth and development, phytohormones, and stress responses (Figure 6A). In the promoter regions of all seven *PoWOX* genes identified in the genome, the number of *cis*-elements related to phytohormone response was higher than the other two categories (Figure 6B). Moreover, *PoWUS* exhibited an extremely large number of TGA-elements associated with auxin responsiveness (46) and WRE3 associated with wound responsiveness (46). However, only one TCCC-motif associated with light responsiveness, which is classified into plant growth and development. This indicates the function of *PoWUS* in phytohormones and stress response. Among the *cis*-elements associated with phytohormones, the number of *cis*-regulatory elements linked to abscisic acid, MeJA, and salicylic acid was higher than other phytohormones. Among the *cis*-elements associated with development, the *cis*-elements linked to light were more prevalent than other factors. Among the *cis*-elements associated with stress, the *cis*-elements linked to drought were more prevalent than other factors. This indicates the function of *PoWOX* genes in response to abscisic acid, MeJA, salicylic acid, light, and drought.

### 2.7. The Tissue Expression Pattern of PoWOX Genes

To validate the expression profile, the six *PoWOX* genes identified in the transcriptome were selected for qRT-PCR across five tissues (root, stem, leaf, flower, and seed) of *P. ostii*. It was found that they were constitutively expressed in different tissues (Figure 7). *PoWOX4* showed significantly elevated expression in roots, stems, and leaves relative to flowers and seeds, with roots exhibiting the highest expression, followed by leaves. The expression level of *PoWOX5* was significantly higher in roots, leaves, and seeds compared to stems and flowers, with roots showing the highest expression level, and there was no significant difference between the expression levels in leaves and seeds. *PoWOX9* showed markedly higher expression levels in leaves and seeds than in roots, stems, and flowers, and the highest expression level was detected in the leaves, followed by seeds, with no significant difference observed among the other three tissues. *PoWOX11* is expressed at a significantly higher level in the roots than in the other four tissues. *PoWOX13a* is significantly expressed at the highest levels in the flowers and leaves, and at the lowest levels in seeds. *PoWOX13b* is significantly expressed at the highest levels in roots, then in stems and leaves, and is lowest in flowers and seeds. Among the six *PoWOX* genes, *PoWOX4*, *PoWOX5*, *PoWOX11*, and *PoWOX13b* were all expressed at the highest levels in roots, indicating their role in root growth and development.

### 2.8. The Expression Analysis of PoWOX Genes in Roots

To further examine the expression of *PoWOX* genes in roots, the expression data of nine sequences annotated as *WUSCHEL*-related genes and capable of encoding amino acids were downloaded from the transcriptome database of *P. ostii*. The transcriptome database contained RNA-seq data of roots at four critical stages: the early stage of root primordium formation (Gmfq), the root primordium formation stage (Gmf), the root protrusion stage (Gtq), and the root outgrowth stage (Gzc) [45]. A heatmap was drawn after data standardization using TBtools (Figure 8A). Most of the nine sequences were highly expressed at Gmfq, except for the two sequences clustered in the WOX13 clade, including CL4464.Contig2, which was highly expressed at Gmf, and Unigene32342, which was highly expressed at Gzc. To confirm the expression levels of the *PoWOX* genes identified from transcriptome data, quantitative real-time polymerase chain reaction (qRT-PCR) was conducted on roots at Gmfq, Gmf, Gtq, and Gzc (Figure 8B). All six *PoWOX* genes were significantly expressed at the highest levels at Gmfq. The expression levels of *PoWOX4*, *PoWOX5*, *PoWOX9*, *PoWOX11*, and *PoWOX13a* decreased gradually with the development of roots, while *PoWOX13b* was expressed the lowest at Gtq, showing a similar trend to the transcriptome expression data. The expression analysis of *PoWOX* genes in roots suggests that *PoWOX4*, *PoWOX5*, *PoWOX9*, *PoWOX11*, *PoWOX13a*, and *PoWOX13b* may play a vital role in the early stage of root primordium formation.

## 3. Discussion

The WOX family is one of the most conserved transcription factor families in plants, playing crucial roles in the normal growth and development of plants. Due to the important role of the WOX family, research on it has been extensive and thorough in plants. However, there has been very little research on the WOX family of tree peony. Only Xia et al. [43] have cloned four *PoWOX* genes, *PoWOX1* (WEX29572), *PoWOX4* (WEX29575), *PoWOX11* (WEX29574), and *PoWOX13* (WEX29573), from the genome and preliminary somatic embryo transcriptome data of *P. ostii*, and then analyzed the expression pattern, subcellular localization, and protein interactions. In this study, we screened both the genome [44] and root transcriptome data [45] of *P. ostii* and identified 10 WOX members, named as *PoWUS*, *PoWOX1a*, *PoWOX1b*, *PoWOX3*, *PoWOX4*, *PoWOX5*, *PoWOX9*, *PoWOX11*, *PoWOX13a*, and *PoWOX13b*. Compared with the deduced amino acid sequences of *PoWOX* genes in Xia et al. [43], PoWOX1a and PoWOX11 are longer and almost completely cover PoWOX1 (WEX29572) and PoWOX11 (WEX29574), and PoWOX4 and PoWOX13b have only one amino acid difference with PoWOX4 (WEX29575) and PoWOX13 (WEX29573), respectively.

### 3.1. Phylogenetic Analysis of PoWOX Proteins

The evolution of the WOX family is closely related to morphological innovations in plant species differentiation [3,46,47]. WOX proteins are divided into three major clades: the WUS clade which contains WUS and WOX1-7, the intermediate clade which includes WOX8, 9, and 11–12, and the ancient clade which includes WOX10, and 13–14 [3,46,48,49,50]. Furthermore, Li et al. [50] and Zeng et al. [51] subdivided WOX proteins into nine subclades, comprising WUS, WOX1/6, WOX2, WOX3, WOX4, and WOX5/7 in the WUS clade, WOX8/9 and WOX11/12 in the intermediate clade, and WOX10/13/14 in the ancient clade. Among the three major clades, the WOX member numbers of the intermediate and WUS clades were higher than the ancient clade, and the WUS clade had the most WOX members. In this study, the WOX proteins of *P. ostii*, *A. thaliana*, and *P. trichocarpa* were classified into three major branches originating from the root. Among the ten members of the PoWOX proteins, PoWUS (Pos.gene14786), PoWOX1a (Pos.gene9686), PoWOX1b (Pos.gene27056), PoWOX3 (Pos.gene36673), PoWOX4 (Pos.gene10408), and PoWOX5 (Pos.gene81040) were classified in the WUS clade, PoWOX13a (Pos.gene46558) and PoWOX13b (Unigene39311) were classified in the ancient clade, and PoWOX9 (Unigene55372) and PoWOX11 (CL8518.Contig2) were clustered in the intermediate clade. These findings are consistent with previous studies.

In the kingdom of plants, *WOX* genes have been found in eudicots, monocots, bryophytes, lycopodiophytes, and unicellular green algae (*O. lucimarinus*, *O. tauri*), but they have not been found in red algae (*Cyanodioschyzon merolae*), which evolved earlier than green algae, or in unicellular green algae (*Chlamydomonas reinhardtii*). This indicates that the *WOX* genes may have originated from green algae and were successively lost in the *Chlamydomonas* lineage [52,53]. Furthermore, the ancient clade includes WOXs that are evolutionarily conserved from algae to higher plants, the intermediate clade includes members found in vascular plants, and the WUS clade exclusively consists of members from spermatophytes [53,54]. The *WOX* genes in the ancient clade emerged prior to the divergence of monocots and dicots, undergoing minimal variations as the categories differentiated [50]. In this study, a phylogenetic tree was constructed with a total of 138 WOX proteins from 17 species, including algae (3), bryophytes (2), lycophytes (1), pteridophytes (2), gymnosperms (2), basal magnoliophyta (1), monocots (2), and eudicots (4). We found that most WOX proteins were clustered in the WUS clade and only existed in angiosperms, consistent with Xia et al. [43]. The ancient clade covered all plant species, while the intermediate clade only clustered two WOX proteins of pteridophyte plants. This indicates that the evolutionary trend in WOX members aligns with the evolutionary trend in plants, and the number of WOX members expanded as plants evolved. 

### 3.2. Sequence Analysis of PoWOX Proteins

WOX proteins, which belong to the homeobox transcription factor superfamily, are distinguished by a conserved homeodomain [3]. A typical homeodomain contains 60 amino acid residues, forming a helix-loop-helix-turn-helix structure capable of binding to specific DNA sequences [1,3]. The ten PoWOX protein members identified in this study all contained a highly conserved homeodomain, although their positions varied, consistent with the results of Xia et al. [43]. In addition, the C-terminus sequences of the homeodomain in different WOX proteins varied. For instance, PoWUS, PoWOX1a, PoWOX1b, PoWOX3, PoWOX4, and PoWOX5, which clustered in the WUS clade, had the sequence NVFYWFQNH. PoWOX9 and PoWOX11, clustered in the intermediate clade, had the sequence NVFYWFQNR. PoWOX13a and PoWOX13b, clustered in the ancient clade, had the sequence NVYNWFQNR. These results are consistent with Xia et al. [43], Nardmann and Werr [46], and Zeng et al. [51]. The distribution of conserved amino acid sites in the homeodomain of WOX proteins was remarkably similar among rice, sorghum, maize, *Arabidopsis*, and poplar [48]. There are highly conserved amino acid residues in the homeodomain of WOX proteins, including Q, L, and Y in helix 1, as well as I, V, W, F, N, K, R, and R in helix 3 [1]. In this study, we identified additional conserved amino acid sites in the homeodomain of PoWOX proteins, including R, W, and P in helix 1; G in the loop; P, I, I, and L in helix 2; G in the turn; and N and Q in helix 3. These findings are consistent with the results of Xia et al. [43], Zhang et al. [48], and Li et al. [50].

The homeodomain of WOX proteins is highly conserved, but the sequences of other motifs vary significantly [48]. Haecker et al. [2] found that all the WOX members had a WUS-box motif, except for WOX13. van der Graaff et al. [3] suggested that the WUS-box motif is only present in the members of the WUS clade. The WUS-box motif, with a basic structure of T-L-X-L-F-P-X-X (where X could represent any amino acid residue), is a unique functional domain found in the WOX proteins. The WUS-box motif was an important region for inhibitory activity [49]. Strictly, the WUS-box motif is located at the C-terminus of members in the WUS clade, and the WUS-box motif sequence is T-L-[DEQP]-L-F-P-[GITVL]-[GSKNTCV]. In addition to the homeodomain, a WUS-box motif was found at the C-terminus of PoWOX proteins, which clustered in the WUS clade, consistent with previous studies [2,48,49,50]. In this study, the WUS-box motif sequence of PoWOX proteins was T-L-[EQ]-L-F-P-L-[EHN]. The analysis of MEME revealed that PoWOX proteins contain five conserved motifs. Among them, Motif 1 and Motif 2, corresponding to Motif 1 and Motif 2 in Xia et al. [43], are two fragments of the homeodomain. Meanwhile, Motif 3 (corresponding to Motif 4 in Xia et al. [43]), Motif 4 (corresponding to Motif 3 in Xia et al. [43]), and Motif 5 (corresponding to Motif 5 in Xia et al. [43]) represent the characteristic motifs of three distinct clades. *PoWOX* genes that clustered in the same clade exhibited similar motif arrangements, indicating potential functional similarity among them.

### 3.3. Expression Analysis of PoWOX Genes

The gene expression pattern is closely related to its functions. *AtWOX4* is predominantly expressed in the vascular cambium of *Arabidopsis*, promoting the division and differentiation of stem cells [21]. *OsWOX4* is expressed at significantly higher levels in tender tissues such as buds, roots, and immature seeds [55]. *AtWOX5* and *OsWOX5* (*QHB*) are specifically expressed in the quiescent center (QC) cells of the root meristem in *Arabidopsis* and rice, respectively, to regulate the maintenance of stem cells [2,56,57]. *ZmWOX5B* is expressed not only in the root meristem QC cells of maize but also in the vascular cambial cells [58]. *AtWOX9* is expressed in the basal daughter cell descendants to regulate early embryonic patterning [18]. *WOX9* is abundantly expressed in inflorescence tissue, playing a role in the regulation of inflorescence architecture [59]. Both overexpression and knockout of *NsWOX9* lead to significant leaf blade distortions, highlighting its crucial role in leaf blade development [60]. *AtWOX11* is expressed primarily in the root apical meristem and root primary xylem cells, promoting *in vitro* root development and callus formation [61]. *OsWOX11* is mainly expressed during the initiation and development of crown roots in rice, and the expression can respond to the induction of auxin and cytokinin [62]. *AtWOX13* is expressed in roots and inflorescences, playing a role in lateral root development and floral organ formation [31]. In various tissues of peony, *PoWOX1* is highly expressed in leaves, seeds, and callus, *PoWOX4* is predominantly expressed in callus, and *PoWOX11* is primarily expressed in seeds, whereas *PoWOX13* is highly expressed in seeds and roots [43]. In this study, six *PoWOX* genes were expressed constitutively in the roots, stems, leaves, flowers, and seeds of *P. ostii*. Among them, *PoWOX4*, *PoWOX5*, *PoWOX11*, and *PoWOX13b* exhibited the highest expression levels in roots, particularly during the early stage of root primordium formation, suggesting that these genes may play a significant role in root growth and development, particularly in the formation of root primordium. The difference from Xia et al. [43] may have been due to the materials. The tissues used in Xia et al. [43] were collected from tissue-cultured seedlings, while our tissues were collected from plants grown in the field, even though the expression of *PoWOX* genes at various stages of somatic embryos indicates that they may be expressed in the stem apical meristem [43], indirectly supporting our results.

### 3.4. The Relationship of PoWOX Genes with Adventitious Roots

Adventitious rooting in test tube plantlets is crucial for the success of micropropagation. The first cell division that led to the formation of adventitious root primordia in poplar takes place in the cambium [63], highlighting the essential role of the cambium in adventitious rooting. *AtWOX4*, which is mainly involved in the formation and development of vascular tissues in shoot apical meristem and root apical meristem, is an important regulatory factor for cambial stem cells. The TDIF-TDR-WOX4 signaling pathway is crucial in maintaining the vascular meristem during secondary growth [22,23,57,64]. *OsWOX4* directly regulates the transcription of *OsAUX1* by binding to the promoter, playing a negative regulatory role in the elongation of the primary root [55]. *AtWOX5* regulates the maintenance of stem cells in the root apical meristem with *SCR*, *SHR*, and *PLT* [65,66], and also maintains the maximum level of auxin by altering the polarity distribution of auxin in the root apical meristem [67]. *MtWOX5* also regulates the formation of the root apical meristem in conjunction with *MtPLT* and *MtBBM1* [68]. *WOX11/12* upregulated the expression of *LBD16* and *LBD29* by directly binding to their promoters [61,69]. Additionally, *WOX11/12* could also activate the transcription of *WOX5/7* by directly binding to the promoter [30]. The rooting pathway mediated by *AtWOX11* helps to form adventitious roots from leaf explants or adventitious lateral roots from primary roots [70,71]. Furthermore, *AtWOX11* is involved in the transition of vascular cambium initiating cells from xylem/phloem mother cells to new lateral root primordium founding cells [72]. *OsWOX11* recruits ADA2-GCN5 to regulate cell proliferation and stem cell maintenance in the root meristem through histone acetylation and deacetylation [73]. *AtWOX13*, which is dynamically expressed during the initiation and development of primary and lateral roots, promotes the initiation of primordial and lateral roots [31], and it plays a critical role in the cell fate specification of callus [74]. *EupWOX13* is the key regulatory gene for the formation of adventitious roots, adventitious lateral roots, and the elongation of adventitious roots [75]. Considering the synthetic analysis of previous studies and the current study, it is suggested that *PoWOX4*, *PoWOX5*, *PoWOX11*, and *PoWOX13b* may play pivotal roles in the initiation and growth of adventitious roots in *P. ostii*.

## 4. Materials and Methods

### 4.1. Plant Materials

Plant materials for this study were *P. ostii*. The plants of *P. ostii* were cultivated in the fields of Henan Agricultural University in Zhengzhou, China. Plant tissues, including roots, stems, leaves, flowers, and seeds, were collected for the extraction of total RNA. In addition, samples at four crucial stages of adventitious root development, namely Gmfq, Gmf, Gtq, and Gzc, were also collected for total RNA extraction [45]. Plant samples were collected in triplicate, immediately frozen in liquid nitrogen, and then stored in an ultra-low temperature (−80 °C) freezer.

### 4.2. RNA Extraction and cDNA Synthesis

Total RNA was isolated using the cetyltrimethylammonium bromide (CTAB) method. Subsequently, the concentration was verified by a spectrophotometer, while the purity and integrity were assessed through 1% agarose gel electrophoresis. Then, DNase I was added to the total RNA to remove any residual genomic DNA, followed by the use of an Evo M-MLV RT Mix kit (AG, Changsha, China) to synthesize the cDNA. 

### 4.3. Identification of WOX Family Members

The identification of WOX family members in *P. ostii* was performed using the assembled genome data downloaded from a public website (https://db.cngb.org/search/project/CNP0003098/, accessed on 29 March 2024) [44] and the RNA-seq data at four stages of root development (https://www.ncbi.nlm.nih.gov/bioproject/PRJNA1041056, accessed on 29 March 2024) from our laboratory [45]. The WOX protein sequences of *A. thaliana* and *P. trichocarpa* were downloaded from the PlantTFDB (https://planttfdb.gao-lab.org/, accessed on 11 April 2024). To identify the WOX family, a local BLASTp search was conducted against the genome and transcriptome databases of *P. ostii* using the *Arabidopsis* and poplar WOX sequences as the query by TBtools (E-value: 1 × 10^−5^). Then, the preliminarily screened sequences were searched using the BLASTP tool on NCBI (https://blast.ncbi.nlm.nih.gov/Blast.cgi, accessed on 26 April 2024). The Batch CD-search tool of NCBI (https://www.ncbi.nlm.nih.gov/Structure/bwrpsb/bwrpsb.cgi) was also used to validate the conserved domains of candidate WOX proteins (accessed on 26 April 2024). In addition, DNAMAN Version 8 was used to align the encoded amino acid sequences. Finally, the sequences without the homeodomain and redundant sequences were eliminated. The multiple alignment was performed using Clustal W with the obtained sequences and the WOX proteins of *A. thaliana* and *P. trichocarpa* (Appendix A). Subsequently, the phylogenetic tree was constructed using the neighbor-joining (NJ) method with 1000 bootstrap replicates in MEGA v7.0. All *PoWOX* genes were then named according to the phylogenetic tree.

### 4.4. Physiological and Biochemical Properties Analysis

The basic physicochemical properties were predicted using the ProtParam tool on ExPASy (http://web.expasy.org/protparam/, accessed on 30 April 2024). The transmembrane helices in the protein were predicted using TMHMM-2.0 (https://services.healthtech.dtu.dk/service.php?TMHMM-2.0, accessed on 30 April 2024). The presence of signal peptides was predicted using SignalP-6.0 (https://services.healthtech.dtu.dk/service.php?SignalP, accessed on 30 April 2024). The subcellular localization was predicted using WoLF PSORT II (https://www.genscript.com/wolf-psort.html?src=leftbar, accessed on 30 April 2024).

### 4.5. Phylogenetic, Motif, and Structure Analysis

To further analyze the evolutionary relationship of the WOX family, 128 WOX proteins from various species, including algae, bryophytes, lycophytes, pteridophytes, gymnosperms, and angiosperms (Appendix A), were downloaded from the PlantTFDB (https://planttfdb.gao-lab.org/, accessed on 11 April 2024). After aligning multiple sequences with Clustal W, the phylogenetic tree of 10 PoWOX and 128 WOX proteins was constructed with the maximum likelihood method and 1000 bootstrap replicates in MEGA v7.0. DNAMAN Version 8 was used to perform the multiple sequence alignment. The conserved motifs of PoWOX proteins were investigated using MEME version 5.5.5 (https://meme-suite.org/meme/; accessed on 10 May 2024). The secondary structure of the PoWOX proteins was analyzed using SOPMA (https://npsa-prabi.ibcp.fr/cgi-bin/npsa_automat.pl?page=npsa_sopma.html, accessed on 13 May 2024). The three-dimensional structure of PoWOX proteins was predicted using SWISS-MODEL (https://swissmodel.expasy.org/interactive, accessed on 13 May 2024).

### 4.6. Chromosomal Localization, Collinearity, and Duplication Analysis

According to the genomic gene annotation file of *P. ostii*, the gene structure and chromosomal localization were analyzed and visualized using TBtools. The genome data and annotation gff3-file of *A. thaliana*, *P. trichocarpa*, and *V. vinifera* were obtained from Ensembl Plants (https://plants.ensembl.org/index.html, accessed on 16 May 2024). The intraspecific and interspecific collinear relationships were displayed by the One Step MCScanX tool in TBtools with default parameters. Subsequently, Advanced Circos and Multiple Synteny Plot were, respectively, utilized for visualization.

### 4.7. Putative Promoter Region Analysis

To analyze the *cis*-acting elements in the promoter of *PoWOX* genes, the 2 kb sequences upstream of the translational start site (TSS) were screened from the genome data using TBtools. *Cis*-elements were predicted using PlantCARE (http://bioinformatics.psb.ugent.be/webtools/plantcare/html/, accessed on 11 May 2024) and visualized with TBtools.

### 4.8. Expression Analysis

The expression data of *PoWOX* genes at four stages (Gmfq, Gmf, Gtq, and Gzc) were downloaded from the root transcriptome database of *P. ostii*. Subsequently, they were graphically represented in a heatmap using TBtools after standardization. The primer pairs for qRT-PCR were designed by Primer Premier 5.0 (Appendix A). The qRT-PCR analysis was conducted on five different tissues (root, stem, leaf, flower, and seed) and four distinct root development stages (Gmfq, Gmf, Gtq, and Gzc) using the SYBR^®^ Green Pro Taq HS Mix kit (AG, Changsha, China). The *β-Tubulin* gene (EF608942) was used as an internal control. The relative expression levels were calculated using the 2^−∆∆Ct^ quantitative method.

## 5. Conclusions

In this study, we screened the genome and transcriptome databases of roots at four crucial stages of *P. ostii* and identified ten WOX members, named *PoWUS*, *PoWOX1a*, *PoWOX1b*, *PoWOX3*, *PoWOX4*, *PoWOX5*, *PoWOX9*, *PoWOX11*, *PoWOX13a*, and *PoWOX13b*. According to the phylogenetic relationships, these WOX members were clustered into three major clades, the WUS clade, intermediate clade, and ancient clade, respectively. The conserved motifs and tertiary structures of PoWOX proteins located in the same clade exhibited higher similarity. The *PoWOX* genes were located on Chr01, Chr02, Chr04, and Chr05. They exhibited interspecific collinear relationships with grape, poplar, and *Arabidopsis* but did not show any intraspecific collinearity. The analysis of *cis*-regulatory elements in the putative promoter regions suggested that *PoWOX* genes play an important role in plant growth and development, phytohormones, and stress responses, particularly in responding to phytohormones. Expression analysis showed that *PoWOX* genes were constitutively expressed in various tissues. *PoWOX4*, *PoWOX5*, *PoWOX11*, and *PoWOX13b* exhibited preferential expressions in roots at the early stage of root primordium formation, suggesting their role in root growth and development. These results will offer a foundation for further study on the role of *PoWOX* genes in the root development of tree peony.

## Figures and Tables

**Figure 1 ijms-25-07668-f001:**
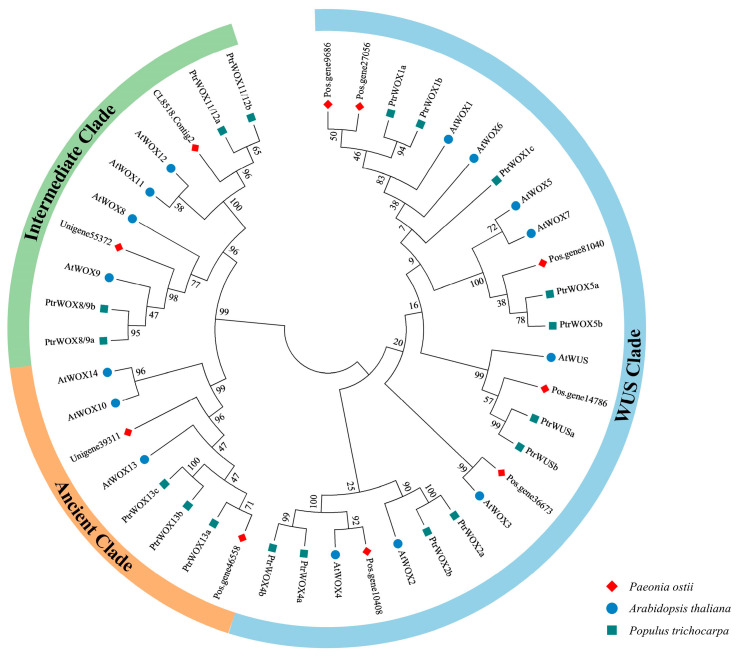
Phylogenetic analysis of WOX proteins in *Paeonia ostii*, *Arabidopsis thaliana*, and *Populus trichocarpa*. The phylogenetic tree was constructed using the candidate WOX proteins in *P. ostii*, along with the WOX proteins from *A. thaliana* (At) and *P. trichocarpa* (Ptr). The tree was generated by using MEGA 7.0 and the neighbor-joining (NJ) method with 1000 bootstrap replicates. The WOX proteins from the three plant species were grouped into three clades. The WUS clade is indicated in blue, the ancient clade is indicated in orange, and the intermediate clade is indicated in green.

**Figure 2 ijms-25-07668-f002:**
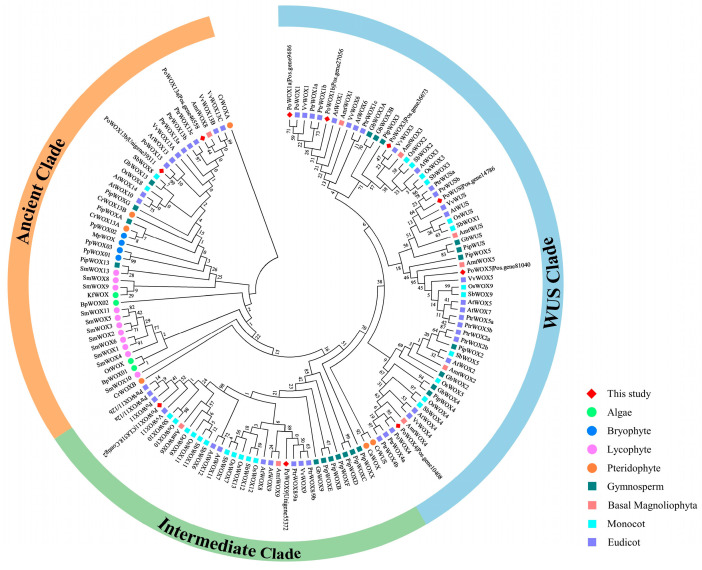
Phylogenetic analysis of WOX proteins in plants. The phylogenetic tree was constructed using a total of 138 WOX proteins, which include 10 PoWOX proteins identified from *P. ostii* and 128 WOX proteins from 17 plant species. These 17 species can be categorized into eight types: algae including *Bathycoccus prasinos* (Bp, 2), *Klebsormidium flaccidum* (Kf, 1), and *Ostreococcus tauri* (Ot, 1); bryophytes including *Marchantia polymorpha* (Mp, 1) and *Physcomitrella patens* (Pp, 3); lycophytes including *Selaginella moellendorffii* (Sm, 11); pteridophytes including *Ceratopteris richardii* (Cr, 5) and *Cyathea australis* (Ca, 1); gymnosperms including *Ginkgo biloba* (Gb, 7) and *Pinus pinaster* (Pip, 14); basal magnoliophyta including *Amborella trichopoda* (Amt, 9); monocots including *Oryza sativa* (Os, 13) and *Sorghum bicolor* (Sb, 12); and eudicots including *P. ostii* (Po, 4), *A. thaliana* (At, 15), *P. trichocarpa* (Ptr. 18), and *Vitis vinifera* (Vv, 11). The phylogenetic tree was generated using MEGA 7.0 with the maximum likelihood method and 1000 bootstrap replicates. The 138 WOX proteins from 17 plant species were grouped into three clades: the WUS clade is indicated in blue, the intermediate clade is indicated in green, and the ancient clade is indicated in orange.

**Figure 3 ijms-25-07668-f003:**
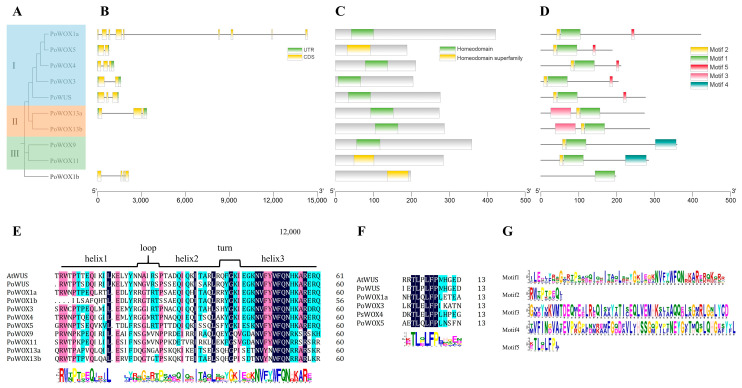
The gene structure, conserved motifs, and sequence alignment of PoWOX proteins. (**A**) The phylogenetic tree of PoWOX proteins, I: the WUS clade, II: the ancient clade, III: the intermediate clade. (**B**) The exon–intron structure of *PoWOX* genes. (**C**) The homeodomain of PoWOX proteins. (**D**) The motif pattern of PoWOX proteins. (**E**) The sequence alignment of the homeodomain in PoWOX proteins, the dark blue color text showed 100% sequence similarity, the pink color text showed 75% sequence similarity, and the light blue color text showed 50% sequence similarity. (**F**) The sequence alignment of the WUS-box motif in PoWOX proteins, the dark blue color text showed 100% sequence similarity, the light blue color text showed 50% sequence similarity. (**G**) The sequence logos of the conserved motifs in PoWOX proteins, the different colors indicated different amino acid residues.

**Figure 4 ijms-25-07668-f004:**
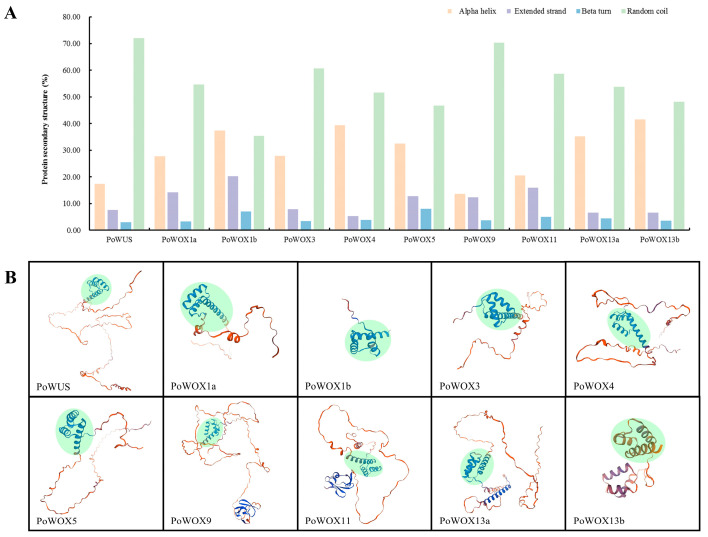
The protein secondary and tertiary structures of PoWOX proteins. (**A**) The secondary structure of PoWOX proteins. (**B**) The tertiary structure of PoWOX proteins. The homeodomain is highlighted in green.

**Figure 5 ijms-25-07668-f005:**
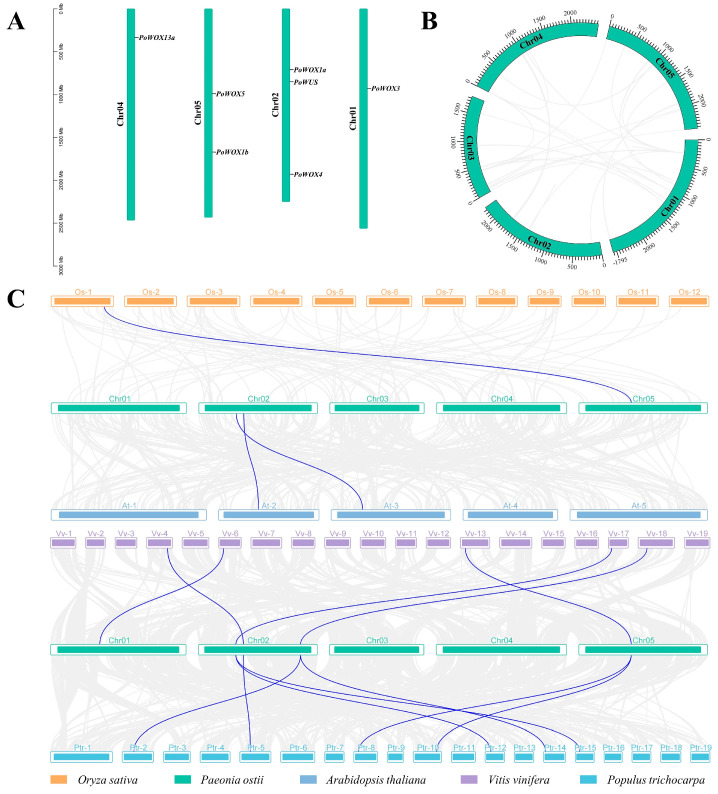
The genome distribution and collinearity relationships of *PoWOX* genes. (**A**) The chromosome location of *PoWOX* genes, the left axis stands for the length of chromosome. (**B**) The intraspecific collinearity relationships of *PoWOX* genes. (**C**) The interspecific collinearity relationships of *WOX* genes in *P. ostii*, *O. sativa*, *A. thaliana*, *P. trichocarpa*, and *V. vinifera*. The blue lines refer to segmental duplicate gene pairs.

**Figure 6 ijms-25-07668-f006:**
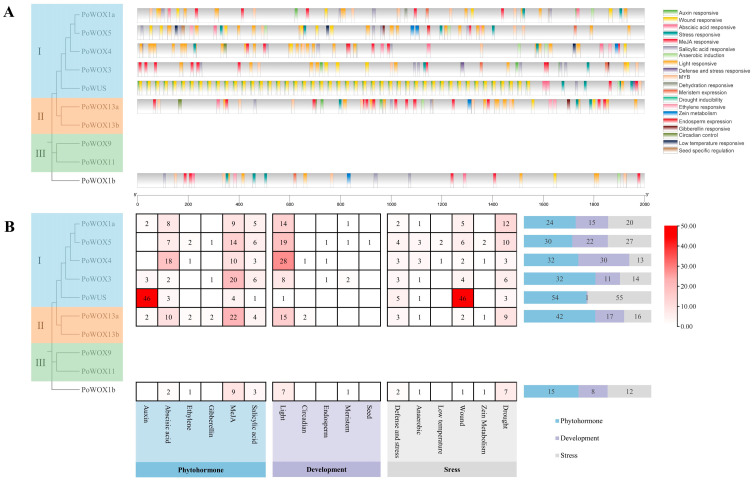
The *cis*-regulatory elements in the promoter of *PoWOX* genes. (**A**) The distribution of *cis*-regulatory elements. (**B**) The statistics of *cis*-regulatory elements.

**Figure 7 ijms-25-07668-f007:**
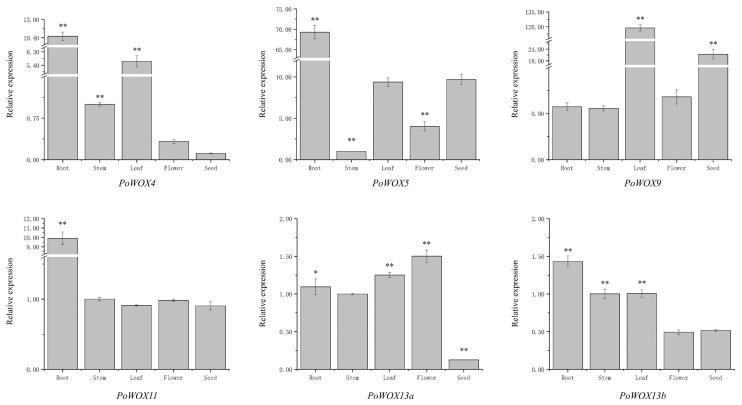
The relative expression of *PoWOX* genes in distinct tissues of *P. ostii*. * indicates significant differences at *p* < 0.05; ** indicates significant differences at *p* < 0.01.

**Figure 8 ijms-25-07668-f008:**
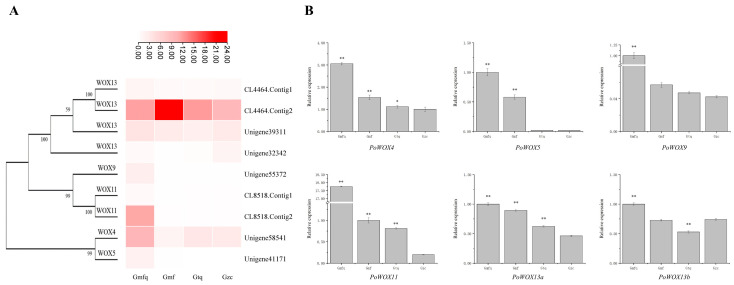
The expression analysis of *PoWOX* genes in the roots of *P. ostii* at various developmental stages. (**A**) The expression levels in the root transcriptome. (**B**) The relative expression in roots at four critical stages. * indicates significant differences at *p* < 0.05; ** indicates significant differences at *p* < 0.01.

**Table 1 ijms-25-07668-t001:** The physiological and biochemical properties of *PoWOX* genes and their deduced amino acid sequences.

Gene Name	*PoWUS*	*PoWOX1a*	*PoWOX1b*	*PoWOX3*	*PoWOX4*	*PoWOX5*	*PoWOX9*	*PoWOX11*	*PoWOX13a*	*PoWOX13b*
Gene ID	Pos.gene14786	Pos.gene9686	Pos.gene27056	Pos.gene36673	Pos.gene10408	Pos.gene81040	Unigene55372	CL8518.Contig2	Pos.gene46558	Unigene39311
GenBank Accession	PP978709	PP978710	PP978711	PP978712	PP978713	PP978714	PP978715	PP978716	PP978717	PP978718
CDS length (bp)	831	1269	597	615	636	567	1077	852	822	861
Number of amino acids (aa)	276	422	198	204	211	188	358	283	273	286
Molecular weight (Da)	31,221.97	47,264.16	22,286.59	23,788.08	24,314.62	21,324.07	39,703.43	31,540.35	31,031.63	32,542.32
Theoretical pI	6.32	6.45	5.55	9.08	9.45	8.74	8.60	6.14	5.29	4.97
Asp + Glu	30	55	28	17	26	24	26	26	36	45
Arg + Lys	28	51	25	22	34	27	29	22	27	31
Instability index	62.4	53.37	41.51	70.98	49.13	55.31	54.93	64.41	73.35	62.97
Aliphatic index	42.1	62.39	83.74	57.89	60.52	69.41	66.15	66.43	64.65	69.27
GRAVY	−1.124	−0.828	−0.313	−0.853	−0.915	−0.619	−0.506	−0.287	−0.844	−0.8
Transmembrane helices	0	0	0	0	0	0	0	0	0	0
Signal peptide	No	No	No	No	No	No	No	Yes	No	No
Localization prediction	Nucleus	Nucleus	Nucleus	Nucleus	Nucleus	Nucleus	Nucleus	Nucleus	Nucleus	Nucleus

## Data Availability

The data presented in this study are available in the article and Appendix A.

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
