# Peer review of "Genome-Wide Analysis of the WOX Family and Its Expression Pattern in Root Development of Paeonia ostii"

_ijms, 2024, doi:10.3390/ijms25147668_

Round 1

Reviewer 1 Report

Comments and Suggestions for Authors

This manuscript identifies the WOS gene family in Paeonia ostia using standard bioinformatic analysis methods and provides qPCR evidence to suggest their expression in P. ostia. The manuscript is well-written but there are issues that need to be clarified before it can be published in IJMS. 

1. The introduction is well-written, providing sufficient background and rationale for the study. The references are also properly cited. However, the 35% identity in the iThenticate check suggests the authors should put more effort into their writing to avoid plagiarism issues.

2. The results section is also well described. However, in the phylogenetic analysis of WOS proteins, I cannot find any description of the bootstrap analysis. The authors should employ at least 500-1000 bootstrap repeats and present the bootstrap values on the nodes of the phylogenetic trees.

3. To ensure data availability and transparency of the research, the authors should provide a table listing all the WOS genes/proteins identified in this manuscript and provide the corresponding NCBI accession numbers.

4. The results of this manuscript do not fully support the title. The title indicates 'Their Potential Function in Root Development of Paeonia ostia.' However, I did not find any functional analysis throughout the manuscript. If the authors want to maintain this title, they should provide at least knockdown experiments to support it or modify the title to fit the contents of the manuscript.

Comments on the Quality of English Language

Minor editing of English language required

Author Response

Comments 1: The introduction is well-written, providing sufficient background and rationale for the study. The references are also properly cited. However, the 35% identity in the iThenticate check suggests the authors should put more effort into their writing to avoid plagiarism issues.
Response 1: Thank you for pointing this out. We guarantee absolutely no plagiarism. Due to the use of professional terminology, some level of identity is inevitable. We tried our best to reduce the identity and made some changes in the revised manuscript using the track changes feature of the Microsoft Word processing program.

Comments 2: The results section is also well described. However, in the phylogenetic analysis of WOX proteins, I cannot find any description of the bootstrap analysis. The authors should employ at least 500-1000 bootstrap repeats and present the bootstrap values on the nodes of the phylogenetic trees.
Response 2: Thank you for giving us fruitful suggestions. The phylogenetic tree of 138 WOX proteins in Figure 2 was constructed using the maximum likelihood method with 1000 bootstrap replicates in MEGA v7.0. We apologize for omitting the description of the bootstrap in the manuscript. Therefore, we have added a description of the bootstrap in the Materials and Methods section of the revised manuscript, on page 16, lines 558-559. For the sake of aesthetics, we had not presented the bootstrap values on the nodes of the phylogenetic trees before. This omission is not very professional. Therefore, we have added the bootstrap values on the nodes of the phylogenetic trees in Figure 2, on page 5, line 146.

Comments 3: To ensure data availability and transparency of the research, the authors should provide a table listing all the WOX genes/proteins identified in this manuscript and provide the corresponding NCBI accession numbers.
Response 3: We are thankful for your constructive suggestions. After conducting BLASTp searches against the genome and transcriptome databases of P. ostii using the WOX protein sequences of Arabidopsis thaliana and Populus trichocarpa as query sequences, a total of 18 amino acid sequences were initially identified in Paeonia ostii, including nine sequences from the genome database and nine sequences from the transcriptome database. We listed them in Table S1 and made corresponding modifications to the numbering of the Supplementary Tables in the revised manuscript. After eliminating sequences without typical conserved domains and redundant sequences, we identified 10 WOX family members in P. ostii. The corresponding GenBank accession numbers of the 10 PoWOX genes are provided in Table 1 on Page 3.

Comments 4: The results of this manuscript do not fully support the title. The title indicates 'Their Potential Function in Root Development of Paeonia ostii.' However, I did not find any functional analysis throughout the manuscript. If the authors want to maintain this title, they should provide at least knockdown experiments to support it or modify the title to fit the contents of the manuscript.
Response 4: Thank you for pointing this out. We agree with this comment. Therefore, we have modified the title to ‘Genome-Wide Analysis of WOX Family and Their Expression Pattern in Root Development of Paeonia ostii’ on page 1, lines 2-4 of the revised manuscript.

Reviewer 2 Report

Comments and Suggestions for Authors

The manuscript entitled “Genome-Wide Analysis of WOX Family and Their Potential Function in Root Development of Paeonia ostia” is a good approach towards understanding WOX gene family in Paeonia ostia. Though there is a another study available on the same plant and on the same genes https://doi.org/10.3390/horticulturae8030266 , but the study conducted by Lou et al is the base for functional verification of WOX genes. Authors are directed to revise the Abstract as they first defined the WOX and their plant species, and then their results. This is not fluent. Authors are directed to briefly talk about the plant species and then present why WOX gene family is important. Then add your methodology briefly and then present your results. Moreover, the authors are directed to carefully study the mentioned reference paper and discuss it in your discussion too. Good Luck!

Comments on the Quality of English Language

Minor English editing is required.

Author Response

Comments 1: The manuscript entitled “Genome-Wide Analysis of WOX Family and Their Potential Function in Root Development of Paeonia ostii” is a good approach towards understanding WOX gene family in Paeonia ostii. Though there is an another study available on the same plant and on the same genes https://doi.org/10.3390/horticulturae8030266, but the study conducted by Lou et al. is the base for functional verification of WOX genes. Authors are directed to revise the Abstract as they first defined the WOX and their plant species, and then their results. This is not fluent. Authors are directed to briefly talk about the plant species and then present why WOX gene family is important. Then add your methodology briefly and then present your results. Moreover, the authors are directed to carefully study the mentioned reference paper and discuss it in your discussion too. Good Luck!
Response 1: We are thankful for your comments and suggestions. Your suggestions are very constructive. Therefore, we have modified the Abstract in the revised manuscript according to your instructions on page 1, lines 13-31. The mentioned reference paper which were reported by Xia et al. (https://doi.org/10.3390/horticulturae8030266) is reference 43 in our manuscript. We had compared the deduced amino acid sequences of the corresponding PoWOX genes in lines 329-333 of the previous manuscript. After carefully studying the reference paper mentioned, we have elaborated on additional discussions in the Results and Discussion sections of the revised manuscript, on page 4 lines 143-144, page 13 lines 393-394, 404, 410, 417-418, 432-435, and page 14 lines 459-461, 467-471. 

Round 2

Reviewer 1 Report

Comments and Suggestions for Authors

The authors have properly addressed my concern. I think the manuscript is ready to be published.

Comments on the Quality of English Language

Minor editing of English language required